# Coastal Wetland Classification with GF-3 Polarimetric SAR Imagery by Using Object-Oriented Random Forest Algorithm

**DOI:** 10.3390/s21103395

**Published:** 2021-05-13

**Authors:** Xiaotong Zhang, Jia Xu, Yuanyuan Chen, Kang Xu, Dongmei Wang

**Affiliations:** 1School of Earth Sciences and Engineering, Hohai University, Nanjing 211100, China; hhuxj@hhu.edu.cn; 2College of Civil Engineering, Nanjing Forestry University, Nanjing 210037, China; cheny@njfu.edu.cn; 3Jiangsu Province Surveying & Mapping Engineering Institute, Nanjing 210013, China; jschjxk@163.com; 4Jiangsu Provincial Hydraulic Research Institute, Nanjing 210017, China; annzy970826@gmail.com

**Keywords:** coastal wetlands, GF-3, random forest model, feature set optimization, wetland classification

## Abstract

When the use of optical images is not practical due to cloud cover, Synthetic Aperture Radar (SAR) imagery is a preferred alternative for monitoring coastal wetlands because it is unaffected by weather conditions. Polarimetric SAR (PolSAR) enables the detection of different backscattering mechanisms and thus has potential applications in land cover classification. Gaofen-3 (GF-3) is the first Chinese civilian satellite with multi-polarized C-band SAR imaging capability. Coastal wetland classification with GF-3 polarimetric SAR imagery has attracted increased attention in recent years, but it remains challenging. The aim of this study was to classify land cover in coastal wetlands using an object-oriented random forest algorithm on the basis of GF-3 polarimetric SAR imagery. First, a set of 16 commonly used SAR features was extracted. Second, the importance of each SAR feature was calculated, and the optimal polarimetric features were selected for wetland classification by combining random forest (RF) with sequential backward selection (SBS). Finally, the proposed algorithm was utilized to classify different land cover types in the Yancheng Coastal Wetlands. The results show that the most important parameters for wetland classification in this study were Shannon entropy, Span and orientation randomness, combined with features derived from Yamaguchi decomposition, namely, volume scattering, double scattering, surface scattering and helix scattering. When the object-oriented RF classification approach was used with the optimal feature combination, different land cover types in the study area were classified, with an overall accuracy of up to 92%.

## 1. Introduction

Coastal wetlands, which play a significant role in protecting biodiversity, controlling runoff and regulating climate [1,2], are some of the most heavily used and threatened natural systems. Due to the complex ecological conditions of wetlands and the spatial and temporal limitations of field investigations, remote sensing technology has become an important means of wetland mapping and monitoring. Despite the success of optical satellite data in applications such as wetland detection and water level monitoring [3,4,5], optical images are less useful in coastal areas due to cloud cover [6]. Synthetic Aperture Radar (SAR), which provides valuable geophysical parameters over intertidal zones in all-weather and daylight-independent conditions [7,8,9], has emerged as a promising tool for wetland monitoring. In particular, quad-polarized data can provide more details to meet the requirements for wetland classification, and various polarization decomposition methods have been demonstrated to provide abundant polarization features, which improve the classification precision [10,11,12].

Gaofen-3 (GF-3), launched on 10 August 2016, is the first Chinese civilian satellite to be equipped with multi-polarized C-band SAR at the meter-level resolution [13]. The SAR payload can support observations in single-, dual- and quad-polarization modes, and its products can be used in marine environmental monitoring, resource surveys and disaster prevention [14]. In recent years, the advantages of SAR data with high spatial resolution from a variety of satellites, such as RADARSAT-2 [15,16], Sentinel-1 [17] and ALOS-2 [18], have been demonstrated in different applications. However, the usage of GF-3 data is very low, which is likely related to its recent launch date [19]. Coastal wetland classification with GF-3 polarimetric SAR imagery remains a challenge. Currently, only Yellow River Delta has the application with GF-3 data [20], while the Yancheng coastal wetlands have not been discussed. The selection of an appropriate SAR wavelength is one of the vital influential factors for land cover classification [21]. In general, the use of X- and C-bands is preferred for herbaceous wetlands and less dense canopies, while L-band is preferred for woody wetlands, such as swamps and other wetland classes with high biomass [22]. Magaly et al. [23] evaluated the contributions of Radarsat-2 (C-band) and ALOS\PALSAR (L-band) full polarimetric data in characterizing and mapping wetland conditions, and found that the variations in canopy structures were better discriminated with C-band than L-band data, while L-band data was useful in determining the wetness conditions of the ground surface. Other studies have demonstrated that shorter wavelengths, such as C-band and X-band, are better suited for non-forested wetlands vegetation patterns, such as bogs, fens and marshes [24,25]. The Yancheng coastal wetlands, which provide important ecosystem services to local communities, consist primarily of extensive intertidal mudflats, river channels, salt marshes, reed beds and marshy grasslands. Therefore, GF-3 equipped with C-band SAR probably has great potential for coastal wetland mapping in this region, and this article will explore the use of C-band fully polarimetric GF-3 image for wetland classification in the Yancheng coastal development zone of Jiangsu Province, China.

Some studies that used fully polarimetric SAR imagery have noted the influence of different polarimetric scattering features on wetland classification results [26,27]. For example, Millard et al. combined SAR and Lidar data to achieve higher accuracy [28]. Chen et al. integrated 20 polarimetric decomposition algorithms and proposed a feature set optimization method to select the optimal polarimetric features for wetland classification [29]. However, these studies only improved the feature optimization method from a statistical perspective and did not consider the applicability of the features to wetland identification. Other studies have evaluated the importance of different polarization decomposition models in wetland identification. For example, features from the Cloude–Pottier and Freeman–Durden methods were determined to be the best for discriminating land types in wetlands when using RADARSAT-2 data [30,31]. However, they just discussed the influence of features on the wetland classification result according to the overall classification accuracy, while ignoring the influence of features on the classification accuracy of typical wetland vegetation. Especially, Neumann decomposition has been proved useful in crop classification [32], but whether it is suitable for wetland classification has not been discussed.

In this research, we aimed to apply GF-3 data to coastal wetland classification. For this purpose, the eastern coastal wetland of Jiangsu, China, was taken as the study area. The specific objectives of this research were (1) to test the validity of using GF-3 polarimetric SAR imagery to classify coastal wetlands with an object-oriented random forest algorithm; (2) to integrate three frequently used polarimetric decomposition algorithms to extract polarimetric scattering features and propose a feature set optimization method; and (3) to investigate the influence of polarization features on the discrimination of typical land cover types in wetlands, especially different wetland vegetation in coastal tidal flats.

The rest of this paper is structured as follows: Section 2 introduces the study site, reference data and satellite imagery in this research. Section 3 provides a description of the methodology. Section 4 presents the experimental results and discussion. Finally, conclusions and perspectives for future work are outlined in Section 5.

## 2. Study Site and Data

### 2.1. Study Site and Reference Data

This study focused on a typical area in Dafeng District, which is located in the Yancheng coastal development zone of Jiangsu Province, as shown in Figure 1. The climate is temperate and humid throughout the year (annual rainfall 980–1100 mm). The topography is flat, with an elevation of 2.8–3.5 m above sea level. The region has many types of land cover, including *Suaeda salsa*, *Spartina alterniflora*, rice paddies, irrigable land, roads, beach and water (river and fishpond). The scattering characteristics of the river and fishpond in the SAR data used in this study were similar, so these two land cover types were classified into one category (water). Figure 2 shows examples of photographs of the typical land cover types taken during fieldwork.

In order to ensure the reliability of the labeled samples, a field investigation was carried out to determine the type and approximate distribution of typical ground objects. Through the field investigation and visual interpretation, the samples were labeled by field validation. For better visual interpretation, the results of the geographical conditions monitoring and Google Earth images were obtained as auxiliary data. Labeled samples were selected randomly, uniformly and representatively. These data were used for production of ground truth areas in the following text. In each category of samples, the ratio of training to testing samples was 1:5, the numbers of samples points were showed in Table 1.

### 2.2. Satellite Data and Data Processing

Gaofen-3 (GF-3), launched in August 2016, was developed by the China National Space Administration (CNSA), and it is the first Chinese satellite to collect multi-polarized C-band SAR data. Among all SAR satellites around the world, GF-3 has the most imaging modes (12 imaging modes), ranging from single to full polarization, with a resolution of 1 to 500 m [33]. The rich polarization information enables the classification of complex wetland classes. Researchers can request data for download from the China Landsat Data Center (http://data.cresda.com:90/#/home, accessed on 10 May 2021) and Nature Resources Satellite Remote Sensing Cloud Service Platform (http://sasclouds.com/chinese/home/, accessed on 10 May 2021). Currently, only protocol users can download GF-3 data.

The data used in the experiment were C-band fully polarimetric GF-3 images with a resolution of 4.5 m × 5 m in the azimuth direction and range direction. The GF-3 polarimetric SAR data used in this study were acquired on 22 September 2017 and preprocessed with PolSARpro. The data were processed with multi-look processing and filtered by a refined Lee filter to reduce the inherent speckle noise.

Multi-scale segmentation is a bottom-up method that combines adjacent pixels or small segmentation objects, and it is commonly used in object-oriented methods [34]. Pauli images can represent all information contained in PolSAR data, with the three bands in a Pauli image corresponding to the physical scattering mechanism of the ground covers [35]. The Pauli RGB image was analyzed in eCognition for multi-scale segmentation.

## 3. Methodology

### 3.1. Image Segmentation

Multi-scale segmentation (MSR) is one of the most commonly used image segmentation methods. It is a bottom-up region segmentation method, which segments an image into multiple levels [36]. The result of the segmentation is mainly based on three parameters: scale, color–shape and smoothness–compactness. In this study, the optimal segmentation parameters were determined by visual interpretation. The principle of segmentation was to ensure small rivers and roads can be segmented accurately. With groups of experiments, the optimal parameters were 28 pixels (scale), 0.3 (shape) and 0.5 (compactness). The image segmentation was applied on the Pauli RGB image.

### 3.2. Extraction of Polarimetric Scattering Features

Incoherent target decomposition is a common feature extraction method for polarized SAR images. There are two kinds of methods for incoherent target decomposition: decomposition based on eigenvalues and eigenvectors and decomposition based on scattering models. Among the decomposition methods based on eigenvalues and eigenvectors, Cloude–Pottier decomposition [37] is commonly used for wetland classification. Freeman decomposition [38], which is a method based on a scattering model, is also widely used for wetland mapping. However, this three-component scattering decomposition is only feasible under reflection symmetry conditions. In order to solve this problem, Yamaguchi et al. added a helix component to three-component scattering decomposition to account for non-reflection symmetry conditions [39]. Additionally, Neumann decomposition proved to be effective in vegetation extraction [32]. The single-bounce eigenvalue relative difference (SERD) [40], double-bounce eigenvalue relative difference (DERD) [37], Shannon entropy (SE) [41], Span [38] and radar vegetation index (RVI) [42] were also shown to have the ability to classify wetlands [25]. Therefore, these features were used in this study.

As shown in Table 2, 16 polarimetric features were extracted from the above decompositions.

#### 3.2.1. Polarization Features from Decomposition Based on Eigenvalues and Eigenvectors

The effectiveness of Cloude–Pottier decomposition in classifying vegetation has been well verified [43,44]. Cloude–Pottier decomposition analyzes an eigenvector of 3 × 3 coherence matrix T3 and extracts the corresponding eigenvalue. Polarimetric entropy (*H*), polarimetric anisotropy (*A*) and the polarimetric scattering parameter (*α*) can be obtained from Cloude–Pottier decomposition to describe object information:(1)$$H=-\overset{3}{\underset{i=1}{\sum }}p_{i}log_{3}p_{i}$$
(2)$$A=\left(\lambda_{2}-\lambda_{3}\right)/\left(\lambda_{2}+\lambda_{3}\right)$$
(3)$$\alpha =p_{1}\alpha_{1}+p_{2}\alpha_{2}+p_{3}\alpha_{3}$$
where $\lambda_{1}$, $\lambda_{2}$ and $\lambda_{3}$ are the eigenvalues of the coherent matrix; $\alpha_{1}$, $\alpha_{2}$ and $\alpha_{3}$ are the corresponding scattering mechanisms (eigenvectors); and $p_{1}$, $p_{2}$ and $p_{3}$ are the probability of each scattering mechanism.

Other polarization features from Cloude–Pottier decomposition have been proposed: RVI, SERD and DERD were used in the study.

Similar to NDVI in optical imagery, RVI is a vegetation index that can track the dynamics of vegetation growth. RVI is the normalization result of three eigenvalues by Span, and it can be calculated as follows:(4)$$\mathrm{RVI}=\frac{4\lambda_{3}}{\lambda_{1}+\lambda_{2}+\lambda_{3}}$$

The eigenvalues of the coherent matrix are obtained with the reflection symmetry assumption:(5)$$\lambda_{1}^{\prime }=0.5\left({\left|S_{HH}\right|}^{2}+{\left|S_{VV}\right|}^{2}+\sqrt{\left({\left|S_{HH}\right|}^{2}-{\left|S_{VV}\right|}^{2}+4{\left|S_{HH}S_{VV}{}^{*}\right|}^{2}\right)}\right)$$
(6)$$\lambda_{2}^{\prime }=0.5\left({\left|S_{HH}\right|}^{2}+{\left|S_{VV}\right|}^{2}-\sqrt{\left({\left|S_{HH}\right|}^{2}-{\left|S_{VV}\right|}^{2}+4{\left|S_{HH}S_{VV}{}^{*}\right|}^{2}\right)}\right)$$
(7)$$\lambda_{3}^{\prime }=2{\left|S_{HV}\right|}^{2}$$

DERD and SERD are defined as
(8)$$\mathrm{SERD}=\{\begin{array}{c} \frac{\lambda_{1}^{\prime }-\lambda_{3}^{\prime }}{\lambda_{1}^{\prime }+\lambda_{3}^{\prime }},\;\alpha_{1}\leq \frac{\pi }{4}\;or\;\alpha_{2}\geq \frac{\pi }{4}\;\;\;\\ \frac{\lambda_{2}^{\prime }-\lambda_{3}^{\prime }}{\lambda_{2}^{\prime }+\lambda_{3}^{\prime }},\;\alpha_{2}\leq \frac{\pi }{4}\;or\;\alpha_{1}\geq \frac{\pi }{4} \end{array}$$
(9)$$\mathrm{DERD}=\{\begin{array}{c} \frac{\lambda_{2}^{\prime }-\lambda_{3}^{\prime }}{\lambda_{2}^{\prime }+\lambda_{3}^{\prime }},\;\alpha_{1}\leq \frac{\pi }{4}\;or\;\alpha_{2}\geq \frac{\pi }{4}\;\;\;\\ \frac{\lambda_{1}^{\prime }-\lambda_{3}^{\prime }}{\lambda_{1}^{\prime }+\lambda_{3}^{\prime }},\;\alpha_{2}\leq \frac{\pi }{4}\;or\;\alpha_{1}\geq \frac{\pi }{4} \end{array}$$

#### 3.2.2. Polarization Features from Decomposition Based on Scattering Model

Freeman and Durden proposed a three-component scattering decomposition to decompose fully polarimetric SAR data [34]. However, this three-component scattering decomposition is only applicable for reflection symmetry conditions [45]. In order to solve this problem, Yamaguchi et al. added a helix component to the three-component scattering decomposition to account for non-reflection symmetry conditions. Yamaguchi decomposition decomposes the inherent matrix into four components. The inherent matrix can be expressed as follows:(10)$$T=p_{s}T_{s}+p_{d}T_{d}+p_{v}T_{v}+p_{h}T_{h}$$
where *T_s_* is the Bragg surface scattering model; *T_d_* is the Fresnel double-bounce scattering model; *T_v_* is the volume scattering model; *T_h_* is the helix scattering model; and *p_s_*, *p_d_*, *p_v_* and *p_h_* are the corresponding power.

The total power of the polarimetric SAR and Shannon entropy can be obtained as follows:(11)$$Span=p_{s}+p_{d}+p_{v}+p_{h}$$
(12)$$SE=\log\left(\pi^{3}e^{3}\left|T_{3}\right|\right)$$

In 2010, Neumann proposed another model-based incoherent PolSAR decomposition method and introduced a generalized volume scattering model. There are four parameters in Neumann decomposition: the orientation angle of the particle, the orientation randomness, and the magnitude and phase of particle scattering anisotropy. The generalized volume scattering model is defined by the particle shape and the orientation randomness. It is assumed that the volume is a cloud of randomly oriented particles and that the orientation angles of the particles follow a normal distribution. The normalized coherent matrix and the volume scattering model are defined as follows:(13)$$T=\left[\begin{array}{ccc} 1 & \delta & 0\\ \delta^{*} & {\left|\delta \right|}^{2} & 0\\ 0 & 0 & 0 \end{array}\right]$$
(14)$$T_{v}=\int_{-\pi /2}^{\pi /2}R_{3}\left(\psi \right)T_{\delta }R_{3}{\left(\psi \right)}^{T}P\left(\psi \right)$$
where δ is the magnitude of particle scattering anisotropy; ψ is the orientation angle of the particle; $R_{3}\left(\psi \right)$ is the rotation matrix; and P(ψ) is the probability density function of ψ. The orientation randomness τ can be defined with the modified Bessel function of order zero:(15)$$\tau =I_{0}\left(\kappa \right)e^{-\kappa },\;\tau \in \left[0,1\right]$$
where κ is the degree of concentration; and $I_{0}\left(\kappa \right)e^{-\kappa }$ is the modified Bessel function of order zero. The larger the value of τ, the greater the randomness of the particle distribution. Two types of linear models of the coherent matrix can be expressed by the linear distribution of the orientation distribution:(16)$$T_{v}\left(\delta ,\tau \right)=\{\begin{array}{c} \frac{1}{1+{\left|\delta \right|}^{2}}\left[\begin{array}{ccc} 1 & \left(1-\tau \right)\delta & 0\\ \left(1-\tau \right)\delta^{*} & \left(1-\tau \right){\left|\delta \right|}^{2} & 0\\ 0 & 0 & \tau {\left|\delta \right|}^{2} \end{array}\right],\;\tau \leq \frac{1}{2}\;\;\\ \frac{1}{1+{\left|\delta \right|}^{2}}\left[\begin{array}{ccc} 1 & \left(1-\tau \right)\delta & 0\\ \left(1-\tau \right)\delta^{*} & \frac{1}{2}{\left|\delta \right|}^{2} & 0\\ 0 & 0 & \frac{1}{2}{\left|\delta \right|}^{2} \end{array}\right],\tau >\frac{1}{2} \end{array}$$

The coherent matrix can be expressed by scattering coefficients:(17)$$T=\left[\begin{array}{ccc} \frac{{\langle |S_{HH}+S_{VV}|}^{2}\rangle }{2} & \frac{\langle (S_{HH}+S_{VV}){\left(S_{HH}-S_{VV}\right)}^{*}\rangle }{2} & \langle (S_{HH}+S_{VV})S_{HV}{}^{*}\rangle \\ \frac{\langle (S_{HH}-S_{VV}){\left(S_{HH}+S_{VV}\right)}^{*}\rangle }{2} & \frac{{\langle |S_{HH}-S_{VV}|}^{2}\rangle }{2} & \langle (S_{HH}-S_{VV})S_{HV}{}^{*}\rangle \\ \langle S_{HV}{\left(S_{HH}+S_{VV}\right)}^{*}\rangle & \langle S_{HV}{\left(S_{HH}+S_{VV}\right)}^{*}\rangle & \langle 2{\left|S_{HV}\right|}^{2}\rangle \end{array}\right]$$

In this case, the parameters in Neumann decomposition are defined by (11)–(13), and $\Phi_{\delta }$ is the particle scattering anisotropy phase.
(18)$$\left|\delta \right|=\left|\sqrt{\frac{{\langle |S_{HH}-S_{VV}|}^{2}\rangle +4\langle {\left|S_{HV}\right|}^{2}\rangle }{{\langle |S_{HH}+S_{VV}|}^{2}\rangle }}\right|$$
(19)$$\Phi_{\delta }=Arg\left(T_{12}\right)$$
(20)$$\tau =1-\frac{\left|\langle \left(S_{HH}-S_{VV}\right){\left(S_{HH}+S_{VV}\right)}^{*}\rangle \right|}{\left|\delta \right|\times {\langle |S_{HH}+S_{VV}|}^{2}\rangle }$$

### 3.3. Ranking the Importance of Features and Optimizing the Feature Set Based on Random Forest

Random forest is a machine learning model that was developed relatively recently. Random forest is combined by different decision trees [46], which solves the overfitting problem that arises in single decision trees. There are two parameters that could influence the classification accuracy of random forest: the number of decision tree and the number of features, and the latter is the focus of this study. In this study, when the number of decision trees exceeded 800, the variation in accuracy tends to be flat. So, the parameter of tree number was 800. Furthermore, random forest can be combined with methods to calculate feature importance—mean decrease accuracy (MDA) [47] is one of the most commonly used methods. This method is based on random forest and directly measures the influence of each feature on the prediction accuracy of the model by adding noise to every feature and observing the degree by which the model accuracy is reduced. A decrease in prediction accuracy indicates that the feature has an influence on the model. Thus, the more important the feature, the higher the MDA.

The principles of sequential backward selection (SBS) [48] and MDA measurement were employed here to construct an algorithm for evaluating the importance of different polarization features in discriminating wetland vegetation types using GF-3 data. From the results, the optimized feature subset can be obtained. The workflow for coastal wetland classification with GF-3 was showed in Figure 3. The algorithm is as follows:(1)Train the random forest model and assume that there are *N* trees in the random forest. For a single decision tree *i*, calculate the out-of-bag (OOB) error and record it as *errOOB_1_(i)*.(2)Randomly scramble the values of feature *f* in the OOB data of decision tree *i* and calculate the OOB error. The OOB error calculated here is recorded as *errOOB_2_(i)*.(3)Calculate difference value between *errOOB_2_* and *errOOB_1_* of decision tree *i*. The MDA can be calculated by weighted average. Calculate the *MDA* of feature *f* as follows:(21)$$MDA\left(f\right)=\frac{1}{N}\overset{N}{\underset{i=1}{\sum }}(errOOB_{2}\left(i\right)-errOOB_{1}\left(i\right))$$(4)Rank the features by their importance values, remove the least important feature from the feature set according to the sequence and calculate the overall accuracy of the new feature set (note that the accuracy is calculated on the basis of 10-fold cross-validation).(5)Iterate until the number of features in the feature set is zero.(6)Calculate the accuracy. The feature set with the highest precision is optimal.

### 3.4. Experimental Flowchart

This study focused on the role of different polarization features in the classification of coastal wetlands in GF-3 images. A random forest model and the MDA method were used in the study. The specific processes in this research are as follows:(1)Filtering and multi-look processing were applied to the original GF-3 data.(2)Sixteen polarization features (Table 2) were obtained by performing the corresponding polarization decompositions.(3)Multi-scale segmentation was applied to a Pauli RGB image, and the data were split into multiple objects.(4)Taking the segmented objects as basic units, the samples were randomly selected.(5)The MDA was measured and used to calculate the feature importance. The MDA of each class in the OOB data was determined to obtain the importance value of each feature in each class.(6)The least important features were successively removed according to the order of feature importance, and the classification accuracy was calculated to determine the optimal feature subset.(7)Data was classified with the optimal feature subset. The classification accuracy was calculated with validation samples.


## 4. Results and Discussion

### 4.1. Importance of Polarization Features for Wetland Classification

Figure 4 shows the MDA of each feature after adding noise to it. The most discriminating features were SE, Y4_Vol, Span and Neu_tau, which had MDA values of 0.247, 0.213, 0.169 and 0.124. The least discriminating features were Neu_psi, Neu_mod and α, the MDA values of which were less than 0.05.

To explore the influence of each feature on classification, we also calculated MDA values for each feature in every category.

The statistical results for non-vegetation are shown in Figure 5a–c. The results clearly show that SERD was the most important feature for the beach, with an MDA of 0.012, followed by SE and Neu_tau, which had little effect, with an MDA of less than 0.01. In the classification of water, Span, SE and Y4_Odd were the most important features, with an MDA of 0.0394, 0.0254 and 0.0247. In the classification of road, H and SE were the most important features; their MDA values were 0.0418 and 0.0416.

As shown in Figure 5d,e, SE and Span were the most effective features for typical wetland vegetation. In the classification of *Suaeda salsa*, the three most important features were Neu_tau, SE and Span; their MDA values were 0.0599, 0.0536 and 0.0462. In the classification of *Spartina alterniflora*, the three most effective features were Span, SE and A, with MDA values of 0.0722, 0.0687 and 0.0677.

As shown in Figure 5f,g, the classification of irrigable land and rice paddy was differently affected by the features. For irrigable land, Y4_Vol, Y4_Dbl and Neu_tau were the most discriminating features, and their MDA values were 0.0522, 0.0386 and 0.0356. For rice paddy, Neu_tau, Span and SE were the three most discriminating features, with an MDA of 0.0628, 0.0607 and 0.0543.

### 4.2. Discussion on Different Polarization Features in Typical Wetland Classes

The experimental results in Section 4.1 demonstrate that SE, Neu_tau and Span are key features that affect wetland classification. Boxplots were used to analyze and understand the mechanism of each feature for each land cover.

As observed in Figure 6a, the value of Neu_tau was lowest for the beach and highest for irrigable land, and the feature distribution intervals of Neu_tau for *Suaeda salsa* and rice paddy differed from those of other classes. Therefore, Neu_tau can effectively distinguish between the above categories. As revealed in Figure 6b, the value of SE was the lowest for water and the highest for *Suaeda salsa*. Furthermore, the SE values were quite different between categories. Thus, SE is highly important in all categories. As shown in the boxplot, the value of Span for water was significantly lower than for the other categories, so Span is particularly important in water classification.

### 4.3. Classification Comparison

Different feature sets were constructed in order to compare their effects. The statistical accuracy was the highest when seven features were used. Therefore, seven features with the highest impact on the classification were selected to build the feature set (FS). Different feature sets were constructed to compare the results of different decomposition methods. Combining Span and features from Cloude–Pottier decomposition was always important in wetland classification [11]. In these experiments, when Span was used with features from Cloude–Pottier decomposition, the feature set H/A/span was the best combination. Similarly, Neu_mod and Neu_pha from Neumann decomposition obtained the best classification accuracy when combined with span. Thus, the feature set Neu_mod/Nue_pha/span was used to evaluate the effectiveness of the Neumann decomposition in wetland classification. The features from Yamaguchi decomposition showed high importance, so a feature set named Y4 was constructed with these features. To evaluate the effect of feature optimization, the original feature set with all 16 features, designated ALL, was tested. To illustrate the effectiveness of the method, feature set ALL also be employed by SVM.

The results are shown in Figure 7. As illustrated in Figure 7b, the feature set H/A/span misclassified rice paddy as *Spartina alterniflora*, and the extraction of *Suaeda salsa* was incomplete. With the use of Neu_tua/Neu_pha/span in Figure 7c, rice paddy and *Ssuaeda salsa* were more complete, but the misclassification of rice paddy and *Spartina alterniflora* was worse. The results for Y4 are shown in Figure 7d, which shows that the misclassification of rice paddy and *Spartina alterniflora* decreased, and the extraction of *Suaeda salsa* and rice paddy was more complete. However, part of irrigable land was misclassified as *Suaeda salsa*. Figure 7e shows the result of using all the features for classification; the results improved, but the rice paddy was fragmentized. Figure 7f shows the result of using all the features by SVM, the misclassification was more serious than Figure 7e. As revealed in Figure 7a, when three key features were added to Y4, the confusion between *Suaeda salsa* and irrigable land decreased, and the rice paddy results improved. The boundaries between classes were clear, and the best effect on classification was obtained. In addition, Figure 7 shows that the mapping result for the beach was unsatisfactory and fragmentized. The reason for this is that the data was captured at 3 a.m., which coincided with the ebb tide. The receding tide covered most of the beach.

In order to further evaluate each feature set, the overall accuracy and kappa coefficient of each result were calculated. The results are shown in Table 3, and the confusion matrixes are shown in Table 4, Table 5, Table 6, Table 7, Table 8 and Table 9.

Table 3 shows that the overall accuracy and kappa coefficient were the lowest when using the feature set H/A /span, with values of 77.80% and 0.7319. When the feature set Neu_tua/Neu_pha/span was used for classification, the overall accuracy and kappa coefficient rose to 82.93% and 0.7962. When features from the Yamaguchi decomposition were used, the overall accuracy and kappa coefficient were improved to 87.94% and 0.859. To some extent, this suggests that Yamaguchi decomposition was more effective than Cloude–Pottier and Neumann decompositions in coastal wetland classification. When all the features were used in the classification, the overall accuracy and kappa coefficient were 89.24% and 0.87. When the feature set was reduced to FS, the overall accuracy and kappa coefficient were the highest (92.86% and 0.914). This illustrates the significance of feature optimization. When the feature set ALL was used in random forest and SVM, the accuracy dropped from 89.94% to 85.57%, and the Kappa coefficient dropped from 0.87 to 0.8269. This shows the superiority of random forest over SVM.

### 4.4. Discussion

With the algorithm constructed in this study, SE, Y4_Vol, Span, Neu_tau, Y4_Dbl, Y4_Odd and Y4_Hlx were found to be the most important parameters for wetland classification. The classification accuracy of this optimized feature set was 92.86%, which is higher than that of the original feature set, the accuracy of which was 89.24%. This suggests that feature redundancy occurs when too many features are included in the feature set. Moreover, features with a low classification ability may even cause noise and affect the result.

In general, SE, Y4_Vol, Span and Neu_tau had the most significant influence on wetland classification. Among these features, Neu_tau was highly effective in distinguishing vegetation, Span could markedly improve the mixture of beach and water, and SE could distinguish each class and improve the classification as a whole.

By adding white noise to each feature, MDA was calculated for every class, and the sequence of importance was obtained for each land cover type. *Suaeda salsa* and *Spartina alterniflora* are typical species of wetland vegetation at the study site. Neu_tau, SE and Span were the most important features for the classification of *Suaeda salsa*, and Span, SE and A were the most discriminating features for *Spartina alterniflora*.

In the three decomposition methods used in the study, the parameters obtained from the Yamaguchi decomposition were the most appropriate for wetland mapping. All of the parameters obtained from Yamaguchi decomposition were highly important for wetland classification. Yamaguchi decomposition was generally effective in classifying the wetland, and the accuracy was 87.94%. Although some parameters from the Neumann and Cloude–Pottier decompositions were correlative, the feature set from Neumann decomposition combined with Span had a higher accuracy than that from the Cloude–Pottier decomposition with Span. This indicates that some features from the Neumann decomposition are more appropriate for wetland mapping than those from the Cloude–Pottier decomposition.

Despite the significant importance of Neu_tau in coastal wetland classification, some features, such as Neu_mod and Neu_pis, were incapable of discriminating between categories. Therefore, feature screening is necessary when using multiple decomposition methods.

To illustrate the effectiveness of the method, the classification results of SVM and random forest are shown in Figure 7, and the accuracy and Kappa coefficients are showed in Table 3, Table 4, Table 5, Table 6, Table 7, Table 8 and Table 9. When the same feature sets were employed for the classification, the results of RF were superior to the result of SVM. To some extent, this demonstrated the effectiveness of the classifier in the method proposed in the paper. Comparing the results of the optimal feature set and the feature set without optimization, the result of FS was superior than ALL. This demonstrated the utility of the proposed feature set optimization method.

The method proposed in this paper is not only applicable to wetland classification, but also has universal applicability in other land cover classifications, such as for forests, urban areas, crops and so on. As is known to all, features are the basis of classification. The result of the selection of features will affect the result of the classification. This paper took wetlands as the study object, and better classification results were obtained from the classifier and feature selection. Therefore, it is reasonable to believe this method can be applied to other land cover classifications as well.

## 5. Conclusions

SAR image has become an important means of wetland research, but studies using GF-3 data for wetland classification are scarce. Furthermore, in feature set optimization, analyzing the influence of features on the basis of only the accuracy of wetland classification cannot meet the needs of this type of research. Therefore, in this research, fully polarimetric GF-3 data were used to study the classification of coastal wetlands, determine the influence of different polarization characteristics on wetland classification and identify typical features. The following conclusions can be drawn on the basis of experiments:(1)GF-3 data provide rich and effective observation information, which can be used as a basis for wetland classification. The classification accuracy in this study reaches 92.86%.(2)SE, Span, Y4_Vol and Neu_Tau play key roles in the classification of the wetland in this study. Among these features, SE improves the accuracy as a whole, and Neu_tau and Span are the most discriminating features for crops.(3)For typical wetland vegetation, the most discriminative features are Neu_tau and SE for *Suaeda salsa* and Span and SE for *Spartina alterniflora*.(4)Compared with the Cloude–Pottier and Neumann decompositions, Yamguchi decomposition is more effective in coastal wetland classification with GF-3 images.

Overall, the findings in this paper demonstrate that the presented feature set optimization method has significant advantages in coastal wetland classification when using fully polarimetric GF-3 data. However, there was some misclassification between rice paddy and *Spartina alterniflora*. Additionally, the feature set used in the study only included polarization features and not objected-oriented shape features, so it did not distinguish between the river and the fishpond. Future studies should employ other objected-oriented features or integrate optical and SAR data to distinguish between these two land cover types.

## Figures and Tables

**Figure 1 sensors-21-03395-f001:**
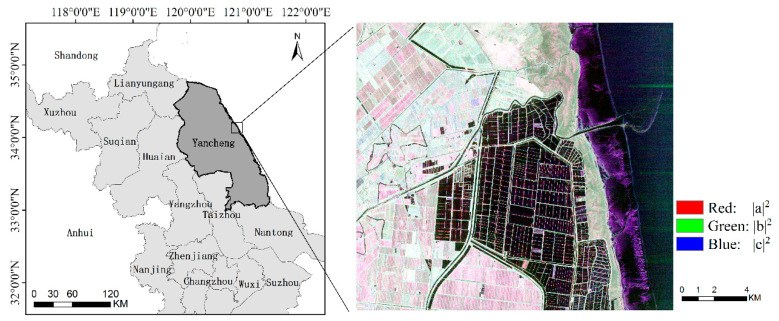
Location and Pauli RGB image acquired on 22 September 2017 by GF-3 of the study site. |a|^2^, |b|^2^ and |c|^2^ are the three components of Pauli decomposition.

**Figure 2 sensors-21-03395-f002:**
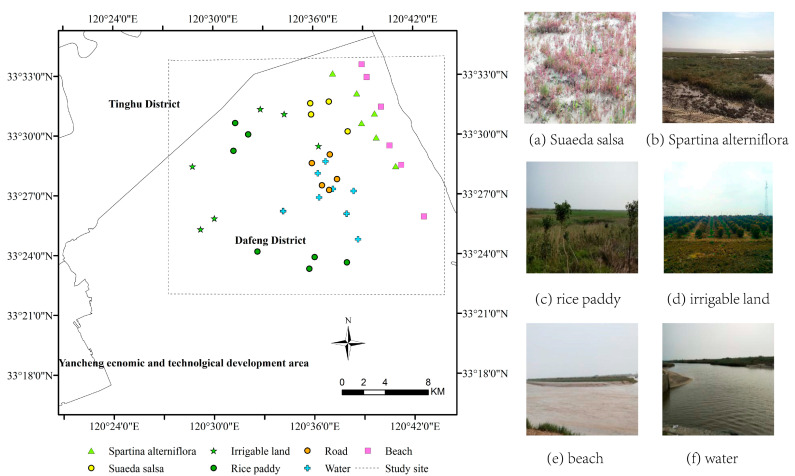
Field survey sites. (**Left**): the distribution of the field sites; (**a**)–(**f**): photos of *Suaeda salsa*, *Spartina alterniflora*, rice paddy, irrigable land, beach and water.

**Figure 3 sensors-21-03395-f003:**
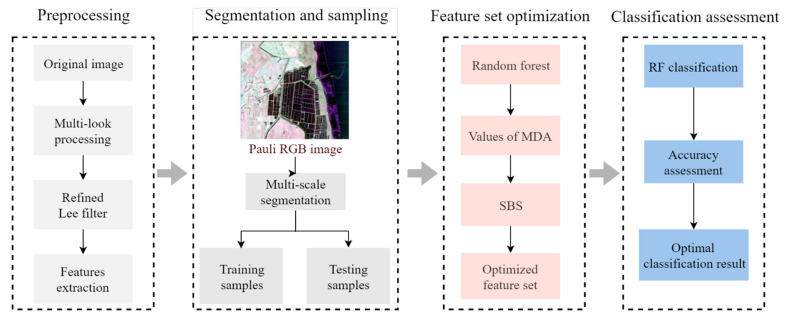
Workflow for coastal wetland classification using GF-3 data.

**Figure 4 sensors-21-03395-f004:**
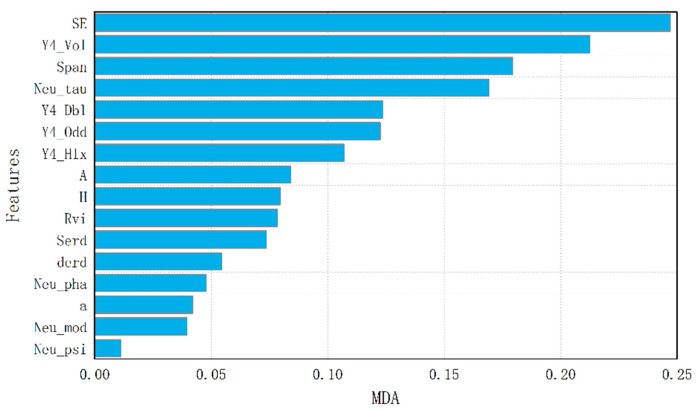
The importance of each polarization feature in wetland classification.

**Figure 5 sensors-21-03395-f005:**
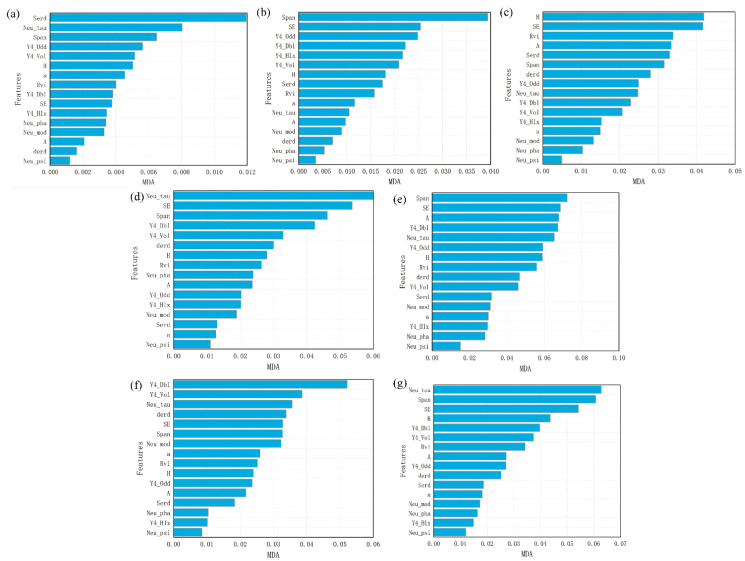
The importance ranking of each feature in the classification of (**a**) beach, (**b**) water, (**c**) road, (**d**) *Suaeda salsa*, (**e**) *Spartina alterniflora*, (**f**) irrigable land and (**g**) rice paddy.

**Figure 6 sensors-21-03395-f006:**
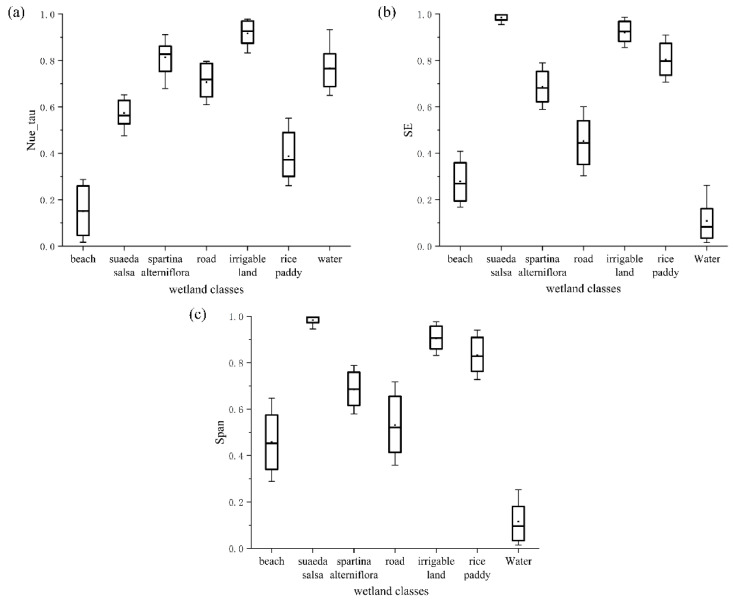
Boxplots of the features in different classes: (**a**) the distribution of Neu_tau among different classes; (**b**) the distribution of SE among different classes; and (**c**) the distribution of Span among different classes.

**Figure 7 sensors-21-03395-f007:**
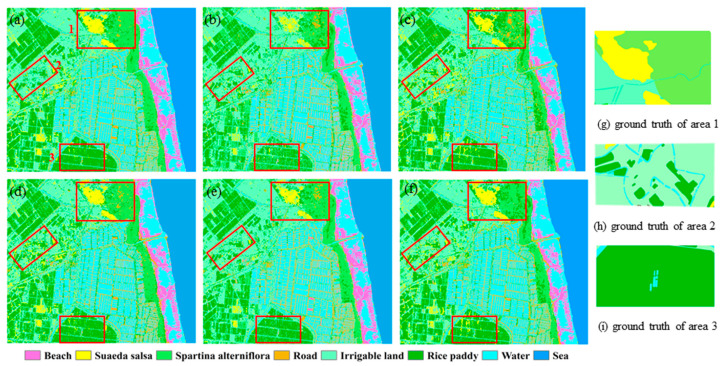
Classification result using different feature sets: (**a**) FS with random forest; (**b**) H/A /span with random forest; (**c**) Neu_tua/Neu_pha/span with random forest; (**d**) Y4 with random forest; (**e**) ALL with random forest; (**f**) ALL with SVM; (**g**) ground truth of area 1; (**h**) ground truth of area 2; and (**i**) ground truth of area 3.

**Table 1 sensors-21-03395-t001:** Sample points setting.

Class	*Suaeda Salsa*	*Spartina Alterniflora*	Rice Paddy	Irrigable Land	Road	Beach	Water
Training Samples	702	2057	1660	1743	384	1618	1232
Testing Samples	3465	10,333	8330	8717	1334	8073	5815

**Table 2 sensors-21-03395-t002:** Acronyms and description of the 16 polarimetric scattering features used in the study.

Polarization Decomposition	Acronym	Physical Meanings
Cloude–Pottier [37]	H	Polarimetric entropy
	A	Polarimetric anisotropy
	a	Polarimetric scattering parameter
Yamaguchi [39]	Y4_Dbl	Double-bounce scattering
	Y4_Vol	Volume scattering
	Y4_Odd	Surface scattering
	Y4_Hlx	Helix scattering
Neumann [32]	Neu_tau	Orientation randomness
	Neu_psi	The average orientation
	Neu_mod	The amplitude of particle anisotropy
	Neu_pha	The phase of particle anisotropy
Other polarization features	SERD [40]	Single-bounce eigenvalue relative difference
	DERD [40]	Double-bounce eigenvalue relative difference
	SE [41]	Shannon Entropy
	Span [38]	The total power
	RVI [42]	Radar vegetation index

**Table 3 sensors-21-03395-t003:** Accuracy of the different feature sets.

Classifier	Feature Sets	Overall Accuracy (%)	Kappa Coefficient
Random forest	FS	92.86	0.914
	H/A/Span	77.80	0.7319
	Neu_tua/Neu_pha/Span	82.93	0.7962
	Y4	87.94	0.859
	ALL	89.24	0.87
SVM	ALL	85.57	0.8269

**Table 4 sensors-21-03395-t004:** Confusion matrix of the classification results from the FS features set using random forest.

Class	Beach	*Suaeda salsa*	*Spartina alterniflora*	Road	Water	Irrigable Land	Rice Paddy	Total	UA (%)
Beach	7682	0	0	3	1	0	0	7686	99.95
*Suaeda salsa*	0	3277	0	0	0	319	196	3792	86.42
*Spartina alterniflora*	40	0	9306	29	10	56	454	9895	94.05
Road	87	0	64	1285	253	0	5	1694	75.86
Water	264	0	0	5	5551	0	0	5820	95.38
Irrigable land	0	112	108	0	0	8228	238	8686	94.73
Rice paddy	0	76	855	2	0	114	7437	8484	87.66
Total	8073	3465	10333	1324	5815	8717	8330	46057	
PA (%)	95.16	94.57	90.06	97.05	95.46	94.39	89.28		

**Table 5 sensors-21-03395-t005:** Confusion matrix of the classification results from the H/A/Span features set using random forest.

Class	Beach	*Suaeda salsa*	*Spartina alterniflora*	Road	Water	Irrigable Land	Rice Paddy	Total	UA (%)
Beach	7257	0	0	235	28	0	0	7520	96.5
*Suaeda salsa*	0	1753	0	0	0	87	108	1948	89.99
*Spartina alterniflora*	27	0	7102	17	9	226	2208	9589	74.06
Road	600	0	194	1055	279	0	192	2320	45.57
Water	189	0	0	17	5498	0	0	5704	96.39
Irrigable land	0	1650	86	0	0	8333	982	11,051	75.40
Rice paddy	0	62	2951	0	1	71	4840	7925	61.07
Total	8073	3465	10333	1324	5815	8717	8330	46057	
PA (%)	89.89	50.59	68.73	79.68	94.55	95.59	58.10		

**Table 6 sensors-21-03395-t006:** Confusion matrix of the classification results from the Neu_tua/Neu_pha/Span features set using random forest.

Class	Beach	*Suaeda salsa*	*Spartina alterniflora*	Road	Water	IRRIGABLE LAND	Rice Paddy	Total	UA (%)
Beach	7165	0	0	5	1	0	0	7171	99.92
*Suaeda salsa*	0	3116	0	1	0	409	453	3979	78.31
*Spartina alterniflora*	61	0	6523	123	1	144	435	7287	89.52
Road	579	0	881	1097	483	14	112	3166	34.65
Water	268	0	0	28	5330	0	0	5626	94.74
Irrigable land	0	94	27	0	0	8022	387	8530	94.04
Rice paddy	0	255	2902	70	0	128	6943	10298	67.42
Total	8073	3465	10333	1324	5815	8717	8330	46057	
PA (%)	88.75	89.93	63.13	82.85	91.66	92.03	83.35		

**Table 7 sensors-21-03395-t007:** Confusion matrix of the classification results from the Y4 features set.

Class	Beach	*Suaeda salsa*	*Spartina alterniflora*	Road	Water	Irrigable Land	Rice Paddy	Total	UA (%)
Beach	7804	0	0	87	13	0	0	7904	98.73
*Suaeda salsa*	0	3113	0	0	239	210	219	3781	82.33
*Spartina alterniflora*	53	0	7942	13	4	124	682	8818	90.07
Road	7	0	1125	1214	288	0	97	2731	44.55
Water	209	0	0	10	5271	0	0	5490	96.01
Irrigable land	0	99	29	0	0	8199	369	8696	94.28
Rice paddy	0	253	1237	0	0	184	6963	8637	80.62
Total	8073	3465	10,333	1324	5815	8717	8330	46,057	
PA (%)	96.67	89.94	76.86	91.69	90.64	94.06	83.59		

**Table 8 sensors-21-03395-t008:** Confusion matrix of the classification results from the ALL features set using random forest.

Class	Beach	*Suaeda salsa*	*Spartina alterniflora*	Road	Water	Irrigable Land	Rice Paddy	Total	UA (%)
Beach	7661	0	0	38	19	0	0	7718	99.26
*Suaeda salsa*	0	2764	0	0	0	50	60	2874	96.17
*Spartina alterniflora*	53	0	9018	8	9	106	1504	10,698	84.3
Road	88	0	151	1264	104	0	52	1659	76.19
Water	272	0	0	14	5682	0	0	5968	95.21
Irrigable land	0	576	112	0	0	8487	481	9656	87.89
Rice paddy	0	125	1053	0	0	74	6232	7485	83.27
Total	8074	3465	10,334	1324	5814	8717	8330	46,058	
PA (%)	94.88	79.77	87.27	95.47	97.73	97.36	74.8		

**Table 9 sensors-21-03395-t009:** Confusion matrix of the classification results from the ALL features set using SVM.

Class	Beach	*Suaeda salsa*	*Spartina alterniflora*	Road	Water	Irrigable Land	Rice Paddy	Total	UA (%)
Beach	7065	0	0	0	0	0	0	7065	100
*Suaeda salsa*	0	3231	0	0	0	156	466	3853	83.86
*Spartina alterniflora*	48	0	7542	17	18	207	732	8564	88.07
Road	380	0	2	1286	611	0	0	2279	56.43
Water	581	0	0	20	5185	0	0	5786	89.61
Irrigable land	0	154	24	0	0	8283	313	8774	94.4
Rice paddy	0	80	2766	1	0	71	6819	9737	70.03
Total	8074	3465	10,334	1324	5814	8717	8330	46,058	
PA (%)	87.5	93.25	72.98	97.13	89.18	95.02	81.86		

## Data Availability

Data was obtained from Nature Resources Satellite Remote Sensing Cloud Service Platform and are available from http://sasclouds.com/chinese/home/ (accessed on 10 May 2021) with the identity of protocol users.

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
