# Peer review of "Coastal Wetland Classification with GF-3 Polarimetric SAR Imagery by Using Object-Oriented Random Forest Algorithm"

_sensors, 2021, doi:10.3390/s21103395_

Round 1

Reviewer 1 Report

This paper proposes the use of the random forest classifier to discriminate coastal wetland areas with respect to other kinds of surfaces observed by a full-polarimetric SAR. In particular, the authors consider several already existing features and select the best ones by means of their corresponding MDA values. The paper is well-written even if it is a bit limited in its novelty that should be better emphasized. However, the results seem to be encouraging.

See that on page 2 the word “RADASAT-2” should be “RADARSAT-2”.

Please, check or better explain eq. (21). I’m not able to see the dependence on the variable “i” of the summation in its argument. Moreover, in (21) “OBB” should be “OOB”.

It seems that the novel contribution of this paper is confined in the use of the MDA values to select the most important features for wetland classification. Is it right? If so, I have some concerns about the novelty of the manuscript that would justify its publication to a journal. Otherwise, please motivate better the novelty of the proposed approaches emphasizing the differences between already existing papers.

Since some features described in section 3 are based on the assumption of the reflection symmetry, it would be useful to the reader to add a sentence or a footnote recalling that the reflection symmetries can be detected as a preliminary step applying the method proposed in:

[1] Detecting Covariance Symmetries in Polarimetric SAR Images. IEEE Trans. on Geoscience and Remote Sensing, January 2017

Beyond the classification accuracy, it would be interesting also to see the confusion matrix in order to better understand with which one each area is erroneously classified.

Reviewer 2 Report

The authors conduct an object-oriented classification using one full-polarization C-band satellite image. 16 polarimetric features obtained by several different decomposition methods were investigated by statistics comparison. Then the most effective combinations were applied to the random forest classification. The best overall accuracy and kappa coefficients were 93% and 0.91, showing high agreement.

However, the lack of several important information reduced the reliability of the study.

The detailed comments are as follows.

Introduction:

P. 2: The art-of-state of the classification for the wetland using polarimetric SAR imagery was mentioned. How about the classification for other lands? What is the difference between the classification of wetland and other land covers?

Study Site and Data

Figure 1. The caption was too simple. Please add the date, the sensor name, the legend of color composite and the scale of the right figure. In addition, could you add the repetitive points of the different land covers shown in Figure 2.

Methodology

P. 7: The multi-scale segmentation was mentioned only here. Please add more detail about how the segmentation was conducted, e.g. the scale parameters. One figure of the segmented image would be helpful.

The same problem of the random forest. The number of training samples and testing samples were selected from the segmentation was missing. The hyperparameters of the RF model were not mentioned.

Results and Discussion

Figure 7: What the red frames mean? Is it possible to make a ground truth for the target area? Only the comparison of the different feature sets could not prove the utility of the proposed method.

Table 2: Please add a confusion matrix to show the accuracy of each land cover.

As mentioned in the previous comment, I don’t understand how different the wetlands and other land covers. Does the proposed method also work in the other land? Please add some discussion about these.

Round 2

Reviewer 1 Report

I have appreciated the effort made by the authors to improve the manuscript and their replies to my previous comments. However, before defitely accept it, I have the following minor comments:

I have understood now eq. (21). However, I suggest writing ??????2(i)−??????1(i) in the summation to emphasize the fact that they change with index i.

As to ref. [40], Please add authors, volume and number.

The paper needs to be proofread because it still contains some errors and hard-to-read sentences. For instance, the sentence “There are two parameters could influence the classification accuracy of random forest:” should be “There are two parameters which could influence the classification accuracy of random forest:”

Reviewer 2 Report

Most of the comments have been revised in the new version.

Author Response

Thank you for your review of our manuscript. If you have any other questions, please feel free to contact us.